# Gene Therapy for Genetic Syndromes: Understanding the Current State to Guide Future Care

**DOI:** 10.3390/biotech13010001

**Published:** 2024-01-03

**Authors:** Marian L. Henderson, Jacob K. Zieba, Xiaopeng Li, Daniel B. Campbell, Michael R. Williams, Daniel L. Vogt, Caleb P. Bupp, Yvonne M. Edgerly, Surender Rajasekaran, Nicholas L. Hartog, Jeremy W. Prokop, Jena M. Krueger

**Affiliations:** 1The Department of Biology, Calvin University, Grand Rapids, MI 49546, USA; hendemar2000@outlook.com; 2Department of Pediatrics and Human Development, College of Human Medicine, Michigan State University, Grand Rapids, MI 48824, USA; ziebajac@msu.edu (J.K.Z.); lixiao@msu.edu (X.L.); campb971@msu.edu (D.B.C.); will3434@msu.edu (M.R.W.); vogtdan2@msu.edu (D.L.V.); caleb.bupp@corewellhealth.org (C.P.B.); surender.rajasekaran@corewellhealth.org (S.R.); nicholas.hartog@corewellhealth.org (N.L.H.); 3Medical Genetics, Corewell Health, Grand Rapids, MI 49503, USA; 4Office of Research, Corewell Health, Grand Rapids, MI 49503, USA; yvonne.edgerly@corewellhealth.org; 5Pediatric Intensive Care Unit, Helen DeVos Children’s Hospital, Corewell Health, Grand Rapids, MI 49503, USA; 6Allergy & Immunology, Corewell Health, Grand Rapids, MI 49503, USA; 7Department of Neurology, Helen DeVos Children’s Hospital, Corewell Health, Grand Rapids, MI 49503, USA

**Keywords:** gene therapy, genetic syndromes, clinical trials

## Abstract

Gene therapy holds promise as a life-changing option for individuals with genetic variants that give rise to disease. FDA-approved gene therapies for Spinal Muscular Atrophy (SMA), cerebral adrenoleukodystrophy, β-Thalassemia, hemophilia A/B, retinal dystrophy, and Duchenne Muscular Dystrophy have generated buzz around the ability to change the course of genetic syndromes. However, this excitement risks over-expansion into areas of genetic disease that may not fit the current state of gene therapy. While in situ (targeted to an area) and ex vivo (removal of cells, delivery, and administration of cells) approaches show promise, they have a limited target ability. Broader in vivo gene therapy trials have shown various continued challenges, including immune response, use of immune suppressants correlating to secondary infections, unknown outcomes of overexpression, and challenges in driving tissue-specific corrections. Viral delivery systems can be associated with adverse outcomes such as hepatotoxicity and lethality if uncontrolled. In some cases, these risks are far outweighed by the potentially lethal syndromes for which these systems are being developed. Therefore, it is critical to evaluate the field of genetic diseases to perform cost–benefit analyses for gene therapy. In this work, we present the current state while setting forth tools and resources to guide informed directions to avoid foreseeable issues in gene therapy that could prevent the field from continued success.

## 1. Introduction

With the discoveries that DNA codes for genes and that a DNA sequence can have variants that increase disease susceptibility, a future was envisioned in which modifying genetic material to reduce disease risk/progression is achievable. Multiple possibilities arose to modify genetic material (Figure 1) [1,2], including taking cells out of the body to correct genetics followed by delivery back to the individual (ex vivo gene therapy), packaging material to make the changes systemically (in vivo gene therapy), or targeting a tissue or cell to be edited (in situ gene therapy). Gene therapy consists of packaging nucleic acids (plasmid, DNA, RNA, antisense oligonucleotides) or gene editing machinery such as clustered regularly interspaced short palindromic repeats—CRISPR- and CRISPR-associated protein 9 (Cas9)—with guide RNA within a particle, often formed by an attenuated virus or nanoparticle, and delivering it to a cell or tissue to modulate a desired gene [3,4,5,6,7]. While animal models showed incredible promise for gene therapy in the 1970s and 1980s, there were early signs of safety risks posed by delivering biomaterials to humans [8].

One of the first human gene therapy clinical trials, completed in 1990 by Rosenberg et al., involved the transfer of tumor-infiltrating lymphocytes modified with a neomycin resistance gene via a retroviral vector to patients with advanced melanoma [9]. The success of this trial provided proof of concept for the clinical application of gene therapy. With that promise of gene therapy, it is rather surprising to follow the complex multiple-decade history of gene therapy setbacks and complications [1]. However, the excitement associated with gene therapy has finally translated into clinical utility within the past few years, with the FDA and other world regulators approving their use, opening the door for correcting or replacing broader disease genetics [2].

Within rare diseases, genomic sequencing has increased to identify pathogenic variants [5,6], which yields an increasing hope of gene therapy to correct the variants. Rare diseases account for USD 997 billion in healthcare costs annually, impacting 15.5 million people within the U.S. [10]. Internationally, the frequency of rare diseases is uncertain due to limitations in diagnosis, but estimates are greater than 100 million individuals. While each rare disease occurs in less than 200,000 individuals (United States) and in 1/2000 births (European Union) [11], more than 5000 unique, rare diseases add up to a considerable fraction of healthcare costs internationally [12]. As international sequencing initiatives have expanded, so has the number of diagnosed individuals for each rare disease, largely contributed to the sharing of flagged genomic variants across borders [13,14,15]. The International Rare Diseases Research Consortium (IRDiRC), founded in 2011, has set forth a critical mission of expanding therapeutics for international usage through integrating international efforts into funding within each country or foundation [16,17]. This international partnership highlights the growing efforts to expand access across borders, which is critical to growing the number of patients with each rare disease to grow the demand and offset drug development costs [18]. The international efforts must continue to translate the United States and European union clinical trials into cross-border initiatives to increase clinical trial implementation for rare diseases [19].

As diagnoses of rare diseases have improved with the implementation of genome sequencing [20,21,22], the knowledge of the exact variant for each individual yields details of how to best treat each case [23,24,25]. If a variant results in loss of function of a protein, it is possible to replace that protein with a functional gene (gene delivery) or remove the cell, followed by CRISPR editing. If a variant causes a gain of function, one can reduce the function using antisense oligonucleotides. Thus, rare diseases are one of the areas where gene therapy holds incredible promise. However, a balance must be maintained between evaluating gene therapy benefits and safety risks to have a sustainable gene therapy ecosystem moving forward. Within this review article, we address the field’s current state in rare diseases and provide insights and guidance to advance the clinical use of gene therapy sustainably and safely. The article consists of an analysis of gene therapy based on publications, funding, status of clinical trials, and approved clinical usages while expanding considerations for additional rare disease genes, immune modulation, cost of therapy, and the need for increased transparency. At the end, the work is concluded through a discussion of the current and future ethical considerations for gene therapy advancement.

## 2. Past and Current Work in Gene Therapy

### 2.1. Publications

The advancements and applications of gene therapy can be reflected in yearly publications (Figure 2). Publications mentioning “gene therapy” date back to the 1970s (1922 total papers) but expanded rapidly in the 1990s (76,314 papers) to the 2000s (317,383 papers) and 2010s (637,126 papers). The number of papers per year seems to have stabilized at the beginning of the 2020s, with 2020 having 88,853 papers, 2021 having 98,207 papers, and 2022 having 99,992 papers. In 2022, the gene therapy papers reflected diverse topics based on a Web of Science analysis. These include general fields like genetic heredity, biochemistry, and pharmacology. More specialized fields such as oncology, immunology, and neurosciences rank the highest in 2022 publications (Figure 2). There are a total of 802,029 papers for “gene therapy” and “Genetic Heredity” over all years, with 25,280 of those articles also containing “Rare Disease.” A similar search within PubMed for “gene therapy” and “rare disease” returns 16,032 papers.

Literature analysis provides valuable insights, especially those of nucleotide delivery systems for studying animal modeling of rare diseases. In the 2000s, a strategy known as morpholino oligonucleotides was widely used in research to knockdown genes in animal models [26]. Building on the toxic nature of oligonucleotides in developmental studies [27], morpholinos were developed to inhibit gene translation using chemical alterations of the oligonucleotide that allow for complementation with the transcript to prevent ribosome engagement [28]. In 2000, these morpholinos were shown to be functional in the knockdown of zebrafish genes during development, mimicking rare disease phenotypes [29]. This novel animal modeling tool progressed with hundreds of papers defining knockdown to phenotype correlations for rare genetic disorders [30]. However, in 2007, the same group that had presented the promise of zebrafish morpholinos showed that the system also regulated the tumor protein p53 (TP53, coded by the *p53* gene) cascade and induced phenotypes independent of the targeted morpholino [31], a finding also shown through small interfering RNA (siRNA) [32] and phosphorothioate-linked DNA [33]. While there are off-target oligonucleotide functions in gene regulation, the tools continue to be used through understanding mechanisms and the growth of control datasets [34,35]. For example, our group has shown morpholino use in zebrafish followed by human mRNA recovery allows for definitive outcomes of human genotype-to-phenotype insights and gene therapy modeling for kidney disease [36]. While these techniques are being phased out with newer CRISPR-based animal modeling [37], they still provide a valuable lesson in considering off-target impacts for delivering nucleic acids. These findings highlight the persistent need for refined knowledge of how foreign nucleotides can impact cellular processes to better predict unexpected, off-target outcomes.

### 2.2. Funding

Similar to publications, funding can establish the trajectory of the gene therapy field. The top funder of worldwide science, the National Institutes of Health (NIH), is experiencing rapid funding growth in “gene therapy,” based on an analysis of NIH reporter. Beginning in 2016, funding mentioning “gene therapy” could be found in the project terms of NIH grants (Figure 3). In 2018, the term could be found in project abstracts, and in 2019 within project titles, with a fast elevation to the USD 8.279 billion in total funding for 2022. The 2022 levels of NIH funding broken down by institutes show the top to be the National Cancer Institute (NCI, USD 1.8 billion), followed by the National Institute of Allergy and Infectious Diseases (NIAID, USD 1.5 billion), National Heart Lung and Blood Institute (NHLBI, USD 885 million), and the National Institute of Aging (NIA, USD 669 million).

The top ten highest funded awards from NIH represent a diversity of institutes and initiatives (Table 1). Many of these awards were for mRNA vaccine programs and testing sites (1ZIATR000437, 1U19AI171421, 1U19AI171443, 1U19AI171110, 1U19AI171954, 1U19AI171292, 1U19AI171403), which primarily reflects the SARS-CoV-2 pandemic response. This mRNA vaccine expansion is likely the most significant factor in the rapid funding investments for gene therapy in 2022. A few of these large projects also reflect the growth of gene therapy within oncology (75N91019D00024–0-759102200019–1, 1U24CA224319) and neurodegeneration (5U01AG059798, 1UF1NS131791, 5R01AG068319, 5U19NS120384).

Further refining NIH investments using a co-search with “rare disease” identified 787 funded awards (Figure 4) with 728 unique project numbers totaling USD 526,396,101. Of these awards, 276 are traditional R01 NIH research awards, summing USD 155,491,503 in research. Additional funding for gene therapy comes from intramural awards (ZIA, 76 awards, USD 109,812,041), contract awards (U54, 63 awards, USD 36,059,350; U01, 47 awards, USD 48,207,117), and small research pilot grants (R21, 57 awards, USD 13,314,891). There is a surprisingly low number amongst these awards of trainee funding, such as K08 clinician scientist awards (24 awards for USD 3,829,993), K23 patient-oriented training (12 awards, USD 2,235,855), F30/F31 predoctoral awards (18 awards, USD 755,681), and F32 postdoctoral awards (3 awards, USD 235,260). As gene therapy is one of the most promising clinical tools, there seems to be a need for elevating targeted training awards.

Based on the titles and the public health relevance statements of “gene therapy” and “rare disease” funded grants, there is a diverse clinical perspective (Figure 4). The mention of genes within the abstracts of the projects also reflects this diverse perspective. From the list of genes, funding is in the areas of neuroscience (*TSC*, *MTOR*, *CLN1*, *CMT1A*, *NF1*), neurodegeneration (*APOE*, *TAU*, *TREM2*), cancer (*RUNX1*, *P53*, *MDM2*, *KRAS*), and cystic fibrosis (*CFTR*). As rare diseases are dispersed between the NIH units, with no primary home that focuses on all rare diseases as a single pathology, it is unsurprising that the funding is spread across different institutes. As gene therapy grows in development, it is critical to consider new cross-NIH initiatives focusing on funding gene therapy advancements, especially those outside of oncology or vaccine designs. As the new ARPA-H (Advanced Research Projects Agency for Health) is established in the United States, gene therapy will likely be a significant component of agency design.

### 2.3. Clinical Trials

The translational advancement of gene therapy is reflected through clinical trials and approved therapies. Since 1990, the field has proliferated, with over 2000 completed clinical trials registered on ClinicalTrials.gov (Figure 5, as of 18 April 2023). The first registered trial returned for “gene therapy” (ClinicalTrials.gov Identifier: NCT00001166) was initiated in 1978, an observational trial of “Gyrate Atrophy of the Choroid and Retina” that focused on genetic determination of disease. In the 1990s, trials began expanding, with fast growth in the 2010s to the 2022 level of 397 trials initiated (Figure 5A). Of the studies, 37% of the total results have been marked as completed, 21% are recruiting, 9% are active but not recruiting, 9% have been terminated, and 5% are not yet recruiting (Figure 5B).

Among the completed studies, cancer was the primary target of most trials, as determined by an analysis of the disease categories provided by ClinicalTrials.gov. Monogenic disorders follow. The FDA breaks human clinical trials into four phases [38]. Phase I trials aim to answer whether a treatment may be given safely, assessing for toxicity in a small population. Phase II trials determine if the treatment is effective at different dosages, such as if adequate protein expression is obtained following gene therapy. Phase III trials compare the new treatment to those already utilized or a placebo, often in a larger population, where it is determined if it will be sent for federal approval as a new therapy. Once approved, interventions are continually monitored for adverse events in phase IV. Rare disease trials, particularly those utilizing gene therapy, often combine phases I and II and utilize a stepwise approach.

Among the clinical trials returned when searching for “gene therapy” and marked as complete, most fall under phase I or II trials (Figure 5C). In addition, most of these were only tested in adults (18 years and older). Viral vectors are the most utilized delivery system in gene therapy clinical trials, the most common being adenoviruses, retroviruses, lentiviruses, and adeno-associated viruses (Figure 5D). Of the adeno-associated viruses, AAV2 and AAV5 were the most selected for use. Plasmid DNA delivery, lipofection, and RNA transfer are the most utilized among nonviral vectors.

As “gene therapy” returns trial data irrelevant to interventions, we further filtered genetic diseases with intervention therapies (Table 2). Multiple disorders have completed phase III trials, including cystic fibrosis, hemophilia B, retinal dystrophy, cerebral adrenoleukodystrophy, Spinal Muscular Atrophy (SMA), and β-Thalassemia. It should be noted that enrollment numbers are minimal for many rare diseases due to the low frequency of disorders within the population. This makes it challenging to build placebo control systems and generate sufficient data for FDA approval processes. These issues suggest the need for thoughtful reconsiderations in gene therapy authorization processes in the future [39] in addition to international cooperation efforts.

Table 3 shows a curated list of phase III trials with gene therapy for rare diseases. Among these nine completed studies, four were for SMA using Onasemnogene Abeparvovec (also known as Zolgensma) for different inclusion criteria (NCT03306277, NCT03461289, NCT03505099, NCT03837184). SMA is characterized by an autosomal recessive dysfunction to exons 7 and 8 of the *SMN1* gene, resulting in progressive spinal cord motor neuron degeneration and muscle atrophy [40]. Type 1 SMA decreases muscle tone so severely that children are never able to sit independently. Without intervention, type 1 SMA patients die of respiratory failure prior to their second birthday. The known genetic mechanisms and the progressive debilitating phenotype have resulted in SMA inclusion in many newborn screenings for early detection before the phenotype manifests [41], making it a compelling target for gene therapy intervention. NCT03306277, known as STR1VE, was the first completed gene therapy phase III study, showing in 22 participants that a single AAV9 cDNA intravenous delivery of the *SMN1* gene (Zolgensma) could prevent the phenotype of SMA type 1 [42]. Of the 22 participants, 3 were withdrawn, with 1 due to an unrelated death and 1 due to an adverse event. Of the patients enrolled, they had an average age of 3.7 months at gene delivery, with half identifying as white and 12 as female. All patients with therapy showed marked clinical improvement and achieved independent sitting at 18 months. Of the 22 individuals, 4 showed signs of respiratory distress, 1 with signs of secondary sepsis, and 2 with hepatic elevated enzymes. Presymptomatic genetically screened *SMN1* variant-positive individuals were assessed for earlier delivery of this therapy (NCT03505099), where all 14 patients had marked clinical improvements [43].

Additional phase III trials have been completed for Choroideremia, cerebral adrenoleukodystrophy, β-Thalassemia, and Leber Hereditary Optic Neuropathy. NCT03496012 showed that a single-dose delivery of an AAV2-encoded *REP1* gene targeted to the eye (in situ) with local injections was able to prevent monogenic inherited retinal dystrophies [44]. NCT01896102 showed the ex vivo delivery of CD34+ stem cells treated with lentiviral encoded *ABCD1* to treat males with cerebral adrenoleukodystrophy [45]. Within that study, there was one reported death, 47% of individuals identified as white, all patients were males, and there were eight events of febrile neutropenia, six with a severe fever, and an extensive list of nonserious adverse events. NCT02906202 and NCT03207009 showed the ex vivo delivery of CD34+ stem cells treated with lentiviral encoded βA-T87Q-Globin gene for β-Thalassemia, with a 91% success rate of individuals showing transfusion independence [46]. Four individuals had adverse events, including one case of thrombocytopenia. NCT03406104 showed the intravitreal delivery (in situ) of the AAV2-encoded *ND4* gene to improve vision in individuals with Leber Hereditary Optic Neuropathy [47]. In summary, it should be noted that SMA therapy is the only completed phase III trial with in vivo intravenous gene therapy results.

NCT00073463 started in 2003, aiming to test 100 participants age 12 or older for aerosolized AAV-encoded *CFTR* for the treatment of cystic fibrosis. While the phase I and II studies for this aerosolized therapy showed safety [48,49], the phase III trial showed no improvement in lung function [50]. The trial was terminated with the last enrolled participant in October 2005.

Below is a description of active trials with posted or published results, focusing on serious adverse responses reported. NCT00999609 used subretinal-injected AAV2-encoded *RPE65* to treat retinal dystrophy in 21 patients, where two of the cases showed adverse drug reactions, and one individual showed convulsions [51]. NCT03370913, NCT03392974, and NCT04323098 showed the use of AAV5-encoded Coagulation Factor VIII infusion in 134 males with hemophilia A, where 22 serious adverse events were reported [52]. NCT03569891 used AAV5-encoded Human Factor IX infusion (Hemgenix, etranacogene dezaparvovec) to treat 67 males with hemophilia B, with five severe events, including acute myocardial infarction, gastrointestinal hemorrhage, pseudarthrosis, and acute kidney injury. In nearly all of the recruiting studies, there is a lack of posted results, meaning until completed, most gene therapy clinical trials lack reported data on adverse events. A commonality of gene therapy studies is the prescreening of antibodies towards the AAV system with no reported issues with immunosuppressive agents.

### 2.4. Approved Therapies

The FDA classifies gene therapy products in combination with cellular therapies within the Office of Tissues and Advanced Therapies, where there are 32 approved licensed products (as of 2 August 2023), 8 of which are gene therapies.

Two therapies have been authorized for SMA treatment: Spinraza and Zolgensma. Spinraza (Nusinersen, Biogen) is an antisense oligonucleotide that targets the *SMN2* gene to alter splicing to recover SMN protein function [53]. The phase III trial (NCT02292537) for Spinraza showed success in preventing SMA in 84 patients, with severe adverse events similar to sham control [54]. It should be noted that Spinraza is delivered intrathecally to the cerebral spinal fluid, and one case of post-lumbar puncture syndrome was noted in the clinical trial. Spinraza requires repeat dosing every four months indefinitely to maintain clinical benefits. Spinraza therapy was submitted to the FDA and approved on 23 December 2016 under a fast-track and orphan drug designation. Zolgensma (Onasemnogene Abeparvovec, Novartis Gene Therapies Inc.) was submitted to the FDA on 1 October 2018 and approved on 24 May 2019, creating an intravenous gene therapy for SMA. Zolgensma is a functioning copy of the full human *SMN1* gene, which codes for the SMN protein that is lacking in SMA patients. Zolgensma currently requires only one dose.

Elevidys (delandistrogene moxeparvovec-rokl, Sarepta Theraputics, Inc.) was submitted to the FDA on 28 September 2022 and approved on 22 June 2023 for the treatment of Duchenne Muscular Dystrophy. Approval was limited to ambulatory patients aged 4–5 years. Elevidys utilizes an adeno-associated viral vector (AAVrh74) to deliver a portion of the dystrophin gene “microdystrophin.”. Sarepta was approved under accelerated status by demonstrating that patients treated with Elevidys had increased microdystrophin expression. It was noted in a published FDA summary memo that the decision for approval went against the recommendations made by the Clinical, Clinical Pharmacology, and Statistics review teams, who did not feel the data submitted showed a definite clinical benefit. Elevidys was approved with the contingency that further clinical trial data would be submitted.

Hemgenix (etranacogene dezaparvovec-drlb, CSL Behring LLC) was submitted to the FDA on 24 March 2022 and approved on 22 November 2022 for the treatment of hemophilia B. Luxturna (voretigene neparvovec-rzyl, Spark Therapeutics Inc.) was submitted to the FDA on 16 May 2017 and approved on 18 December 2017 for the treatment of biallelic *RPE65* mutation-associated retinal dystrophy. Skysona (elivaldogene autotemcel, bluebird bio Inc.) was submitted to the FDA on 18 October 2021 and approved on 16 September 2022 to treat active cerebral adrenoleukodystrophy. Zynteglo (betibeglogene autotemcel, bluebird bio Inc.) was submitted to the FDA on 20 September 2021 and approved on 19 August 2022 to treat ß-Thalassemia. Roctavian (valoctocogene roxaparvovec-rvox) was submitted to the FDA on 23 December 2019 (resubmitted 29 September 2022) and approved on 29 June 2023 to treat severe hemophilia A only in the absence of AAV-5 preexisting antibodies. Vyjuvek (beremagene geperpavec) was submitted to the FDA on 20 June 2022 and approved on 19 May 2023 for the treatment of those >6 months of age with dystrophic epidermolysis bullosa due to COL7A1 variants. It should be noted that Vyjuvek is the first ever approved topical gene therapy and utilizes a herpes simplex virus type 1 (HSV-1) delivery system. HSV-1 is optimal for skin delivery as the virus naturally infects skin cells.

In the case of many of these FDA-approved therapies, their phase III trials continued after their authorizations, with an expectation of progression into phase IV studies.

While gene therapy in cystic fibrosis has had mixed results, it should be noted that small molecule regulators of the CFTR gene have proven that nucleotide delivery is not the only approach to modify gene expression in rare diseases. The FDA approved Elexacaftor–tezacaftor–ivacaftor, also known as triple therapy, which is recommended in patients with at least one copy of Phe508del CFTR variants [55,56]. Cystic fibrosis is an example where strategies outside of gene therapy should be continued in parallel, setting a critical mission that gene therapy trials do not overpower or result in underfunding small-molecule or other therapeutic approaches.

## 3. Biological Considerations

For effective gene therapy, one must confidently identify a causal gene, package that gene into a delivery system expressing the right amount in the right tissue/cell, and replace or repair the molecular mechanism with a measurable phenotype. This must be achieved while avoiding unforeseen biological challenges of viral vectors and overexpression of mRNA within cells. Below, we provide several areas of consideration for expanding gene therapy into additional clinical genetics.

### 3.1. Genetic Syndromes

The OMIM database (https://www.omim.org/) [57] represents a catalog of human genetic conditions. As of April 2023, the database contained >6000 gene-to-disease correlations. These correlations represent 4771 unique human genes on all human chromosomes (Figure 6A). Using the UniProt database of protein annotations [58], it is evident that only a few represent DNA binding factors or have annotated domains like a zinc finger or coiled-coil segment (Figure 6B). A significant portion of these proteins are transmembrane, suggesting they localize to the surface of a cell. Many proteins have catalytic activity, binding sites, and active sites. In some rare and genetic diseases, the active site becomes hyperactive, where inhibitors can ameliorate disease. Most diseases manifest from loss-of-function to protein biology and thus need correctors instead of inhibitors.

Using the Human Protein Atlas (HPA) database [59], it is observed that most of the genes are ubiquitously expressed in human tissues (Figure 6C). At the same time, they have more specificity when annotated based on cell types within each tissue (Figure 6D). This observation suggests that we should not address tissue specificity for each gene but rather cell type specificity, where emerging tools like single-cell transcriptomics are opening new doors for these insights. Of the OMIM genes, 2398 have been knocked out in a mouse model, can be purchased for lab use, and have undergone extensive phenotypic analysis based on the International Mouse Phenotyping Consortium (IMPC, Figure 6E) [60]. A total of 90% (2158/2398) of these genes show at least one observable phenotype altered by removing the gene, many matching the known human conditions, where these animals can serve as a pre-clinical gene therapy testing system.

It should be noted that 341 gene knockouts from the IMPC result in heterogeneous preweaning lethality (incomplete penetrance), and 131 are highly penetrant for lethality. The heterogeneity within phenotypes for genetic diseases represents one of the most considerable challenges in gene therapy; namely, how can one develop clinical trials to know success when phenotypes are not always predictable with our current state of knowledge.

It should be noted that the number of datasets showing gene expression within the HPA has little correlation to the number of altered phenotypes observed in the IMPC (R^2^ of 2 × 10^−5^, Figure 6F). This points to the need for further tools in genotype-to-phenotype predictions that will strengthen our ability to know when and how gene therapies may apply to an individual.

Many gene therapy delivery systems have a limited size of the genetic insert, with most of the OMIM genes within this window (Figure 6G). The largest database of human genetics, ClinVar [61], shows that of these OMIM genes, we have an array of known confident pathogenic variants (Figure 6G). While our pathogenic and likely pathogenic variants usually are significant changes to proteins (frameshift and nonsense variants), the current state of research is challenged by missense genetic changes and whether they confidently result in disease states (Figure 6H). Gene therapy can only be employed in high-confidence situations. Thus, the million plus variants of uncertain significance (VUSs) in OMIM genes would have a low probability of successful clinical trials, primarily if implemented based on newborn screening. This finding highlights that variant characterization remains a significant challenge in gene therapy expansion for genetic syndromes.

### 3.2. Cell and Promoter Specificity

Gene therapy is targeted to cell types based on the vector used to deliver the nucleic acids and sequences that can drive the expression of each gene only within that tissue/cell type, such as a cell-specific promoter element. The control of expression enables each gene to be made into mRNA and protein only in a specific cell type. To minimize the size of expression regulation sequences, promoters rather than enhancers are often used to achieve cell-type specificity [62]. Since the advent of RNA sequencing, there has been an expansion in defining tissue/cell-specific expression. Still, more recently, with techniques such as single-cell RNA sequencing, we are now resolving specificity in the different functional cell types within each tissue. This specificity of expression is critical to controlling many OMIM genes contributing to developmental pathways. The HPA annotation of cell specificity for 75 different cell types shows 1908 different human genes with highly specific expression within one of the cell types (Figure 7). More work is needed to determine which promoter elements may work, independent of cell-type-specific enhancers, for the desired tissue of an OMIM gene being nominated for gene therapy.

It should be noted that the developmental trajectory of many genes makes it challenging to identify when gene therapy will be safe and effective. For example, variants that disrupt the complex developmental process of neural crest cells give rise to multiple, diverse peripheral cells [63] and require more critical reasoning on whether gene therapy can recover the developmental changes. Our work on LRP1-related syndrome [64] highlights the complex multi-phenotype traits associated with neural crest cells that will be difficult to advance gene therapy approaches within the developmental stages, often active in utero.

### 3.3. Variant Location within Proteins

As shown in Figure 6H, many clinically sequenced variants within genes that may benefit from gene therapy fall within sites that are difficult to annotate and thus result in an annotation as a VUS. These are often subtle missense variants within a gene and are the first observance of such variants. Most of these variants have only been identified in a single individual and never observed in the millions of sequenced human genomes completed to date, making it difficult to establish a causal nature of the missense variant [22]. Thus, it has become common that gene therapies are initiated only in individuals that have either a variant that occurs in multiple individuals (often autosomal recessive conditions) or the variant results in a frameshift or nonsense change that removes large chunks of protein observed to be removed in other patients with the disorder. There is a need for characterizing VUSs rapidly using existing data [23,64,65,66,67,68], high-throughput wet lab techniques used in NAA10 characterizations [69], knowledge from paralog proteins such as the work on SOX transcription factors [70], or through crowd-sourcing variant lists to identify matching variant locations and phenotypes as was the case for MED13 [71].

Unique variants within genes with early and penetrant phenotypes matched to other pathogenic cases with similar phenotypes are easier to diagnose and determine a missense variant as pathogenic. This relies on phenotype matching, even if variants are unique to a patient. However, in the case of most progressive disorders (such as neurodegeneration) that are detectable in newborn screening before the phenotype is observed, these missense variants cannot be mapped with confidence, preventing the initiation of gene therapy until a phenotype appears. Therefore, if we anticipate gene therapy to apply to every individual for a gene approved with therapy, we must build more robust tools for interpreting each amino acid within an observed gene.

### 3.4. Gene Isoforms and Common Variants

Among the OMIM genes, each gene has an average of 6.2 protein-coding isoforms. These isoforms represent changes in splicing or transcriptional start sites that can alter the sequence of each protein. It is important to remember that many genes have different isoforms within different tissues and that human variants can result in altered splicing [72]. Previously, we showed how variants could alter gene splicing, such as small GTPases [73], and how alternative transcriptional start sites can change the interpretation of common disease association variants, such as SHROOM3 for chronic kidney disease [36].

The *SMN1* and *SMN2* genes each contain multiple spliced isoforms variably expressed in different human datasets based on the GTEx database [72] (Figure 8A). Each of these different isoforms has splice differences that remove one of three exons, resulting in various-sized proteins of each (Figure 8B). New genomic tools such as GTEx have built correlations between genomic variants within genomes and expression (eQTLs) or splicing (sQTLs) for each gene. Both the *SMN1* and *SMN2* genes have eQTLs and sQTLs that modify the genes (Figure 8C). More importantly, these variants are found enriched within human populations such as Africans/African Americans and remain understudied. Interestingly, both the sQTLs in *SMN1* and *SMN2* are found at the C-terminal region of the genes in similar locations (Figure 8D).

While we highlight the role of variants of *SMN1* and *SMN2*, many human genes have variants that can modify expression levels or splicing [72]. However, most of these variants have remained understudied regarding how to incorporate them into gene therapy approaches. This represents a promising area for further exploration as we develop gene therapies for diverse human populations that are increasingly being studied using population-level genomics such as GTEx.

### 3.5. Risk of Overexpression

In gene therapy, determining and controlling the appropriate protein expression level in cells can be challenging, with uncertain outcomes if the expression is too high. Tools are available to help guide us to potential outcomes of gene overexpression, ranging from additional copies to overexpression in disease states. When determining a gene for therapy, it is critical to observe using data analysis tools if the overexpression could result in any measurable phenotypes. This can include the analysis of ClinGen [74] to determine if there are any known genetic events within humans for dosage sensitivity, specifically the genetic duplication of the gene that results in a measurable phenotype (triplosensitivity). As noted above, eQTLs can also tell us when subtle variants, often noncoding, can result in population-level increases in gene expression. These eQTL variants can be compared to Genome-Wide Association Studies (GWASs) or Phenome-Wide Association Studies (PheWASs) to find when these variants associated with elevated expression can also overlap with a measurable phenotype, taking care to determine the maximum peak overlap of colocalization of the expression and phenotype of the same variant [23].

An example of colocalized variants can be seen in the *NF1* gene, which is emerging as a new potential gene therapy target for Neurofibromatosis [75]. The variant chr17_31326275_T_C (rs9894648) is found in diverse populations with significant known *NF1* eQTLs over multiple tissues and a colocalized signal for the variant to traits such as sex-hormone-binding globulin protein (Figure 9). This suggests that modulation of *NF1* levels in gene therapy could have a resulting perturbation in hormone signaling that could be measured over gene therapy trials to determine if this has clinical utility. We must utilize our massive biological knowledgebases, such as eQTLs and GWASs/PheWASs, to determine non-biased traits that should be measured within clinical trials as a risk of overexpression of a chosen gene.

### 3.6. Delivery Systems

A gene therapy delivery system must reach the targeted cells, evade immune system phagocytosis (depleting therapy), and make a functional protein once in the cell while avoiding lysosomal degradation [4]. Delivery strategies such as lipid-based systems and nanoparticles have little cell specificity for delivery, while viral strategies have more surface receptor specificity and higher risks of immune activation [5,77]. Non-viral delivery systems have seen a recent boost with use in SARS-CoV-2 and other mRNA vaccines, which has increased the hope of applying them to broader gene therapies [78]. Newer biological strategies, such as extracellular vesicles, are also emerging as ways to avoid immune activation [79]. Viral vectors such as adeno-associated viruses (AAVs) have lower immune activation and a limited 4.8 kilobase insert size. In contrast, larger viruses such as herpes simplex virus (HSV) have a larger insert capacity but higher immunogenicity with narrower cell targeting [80]. As many of these viruses are natural sources of infection, some individuals carry antibodies or T-cells that are responsive during gene therapy and must be monitored [81]. Substantial ongoing efforts are therefore aimed at reducing the immunogenicity of viral vectors and functionalizing non-viral vectors to enhance cell-type-specific targeting and effects.

Viral delivery systems are often matched to the cell/tissue type of natural infection, opening the door for engineering opportunities to enhance delivery to tissues without an optimal viral system. While there were significant investments in gene therapy approaches for cystic fibrosis, these therapies struggled to find therapeutic benefits due to difficulty in delivery to the progenitor cells of the lung. Hurdles to AAV gene transfer to airway epithelia for cystic fibrosis include (1) by-passing the mucus to reach the cell surface; (2) binding a receptor at the apical cell surface; (3) endocytosis for cell entry; (4) trafficking to the nucleus; (5) conversion of the single-stranded DNA core to double-stranded DNA followed by concatemerization and/or integration; and (6) achieving therapeutic levels of protein expression. As the current small molecule cystic fibrosis drugs are only recommended for individuals with a delta508 variant, gene therapy is still needed to treat individuals of diverse ancestry not having delta508 [82]. Over the past decade, improvement in the efficiency of AAV targeting of airway epithelia has been achieved by using different serotypes [83,84,85,86,87,88], site-directed mutagenesis modifications of viral capsids [89], and targeted evolution selection [90,91]. Currently, ongoing clinical trials using the AAV vector derived from directed evolutions demonstrate promising safety profiles for treating individuals who are ineligible for or unable to tolerate triple therapy (NCT05248230).

The prevailing hope throughout the gene therapy field is that viral delivery systems studied within each trial will be carried forward into the subsequent development to minimize the risks of gene therapy with delivery system human validation data [92]. Multiple AAV clinical trials have pointed towards hepatic injury risks [93], including cytokine/neutrophil-dependent mechanisms [94]. In animal studies, these risks are contributed to by environmental factors such as obesity and diabetes [95]. As gene therapy progresses in clinical trials and FDA-approved clinical use, we must document risk factors for adverse outcomes to each vector and determine environmental or genetic factors to help identify risks.

## 4. Immune Response

Currently, gene therapy is designed to deliver the desired effect in one dose. However, there is a lack of long-term data on the efficacy of these treatments as the FDA approvals have only been in the past few years [96,97]. As more data are obtained about these therapies, redosing may be necessary. The possibility of redosing poses a challenge to gene therapy vectors [98]. Viral vectors have most of their replication machinery removed to enable them to carry the desired gene. However, the vector still contains surface epitopes that elicit innate and adaptive responses against the virus as the wild-type immune response [99,100]. Usually, producing antibodies or T-cell adaptive responses to viral infections is advantageous to help clear infection and enables future viral detection to provide resistance. However, in the case of viral vectors of gene therapy, it is a significant roadblock, as the antibodies may already be present from similar natural infections, or the first dose may inhibit the efficacy of vector reutilization for future doses of gene therapy [101].

The presence of viral vector antibodies before treatment threatens the future accessibility of gene therapy and increases the risk of adverse events. In 1999, the University of Pennsylvania conducted a clinical trial for an adenovirus serotype 5 (Ad5)-based gene therapy for a rare metabolic disease known as ornithine transcarbamylase (OTC). One of the participants suffered from lethal systemic inflammation four days post-treatment [102]. A recent study by Somanathan et al. (2020) presents data suggesting that preexisting Ad5 antibodies may have contributed to the lethal inflammatory response [103]. Additionally, recent deaths in a pediatric high-dose adeno-associated virus (AAV) gene therapy trial for X-linked myotubular myopathy may have been caused by AAV antibodies and an exaggerated immune response similar to that observed in the OTC trial [104]. As a result of the risk of exaggerated immune response, made evident by these incidents, individuals with pre-existing immunity to specific viral vectors are to be excluded from viral-based gene therapy clinical trials [105].

Levels of pre-existing antibodies for AAVs have been noted to be high enough to reduce the patient inclusion population for clinical trials by almost 50% [106]. The prevalence of these antibodies (seroprevalence) can differ across populations. Some populations have been found to have over 90% pre-existing adenovirus immunity by age 2 [107]. The high prevalence of pre-existing antibodies can biologically limit the accessibility of gene therapies to specific populations and even perpetuate current racial disparities in healthcare accessibility. A recent study by Khatri et al. (2022) found seroprevalence was higher among U.S. racial minorities, specifically Hispanic and African American individuals [108]. Therefore, gene therapies utilizing viral vectors may have decreased efficacy in racial minorities.

Zolgensma highlights the gravity of this issue. Zolgensma uses the AAV9 vector. Khatri et al. (2022) found significantly higher AAV9 seroprevalence among black donors than white donors [108]. However, in their study of the differences in *SMN1* allele frequency in North America among different ethnic groups, Hendrickson et al. (2009) found black individuals to have five times the risk of being a carrier for SMA compared to white individuals [109]. The design of Zolgensma creates the potential for a lack of biological accessibility to one of the populations that could benefit the most from it.

To avoid this issue, gene therapy vectors must be chosen with their target population in mind. The vector utilized should be that which, along with being the most biologically functional and effective to deliver the gene of interest, is accessible to the broadest possible range of populations. Antibody titers can be used to measure pre-existing immunity. Two primary assays have been developed: binding assays that measure the total amount of antibodies (neutralizing and non-neutralizing) and neutralizing assays that only measure neutralizing antibodies [105]. Continued monitoring of global seroprevalence and continued prescreening of trial participants and potential gene therapy patients will be necessary to address the growing challenge of pre-existing immunity to viral vectors.

Research is needed to understand the immune response to viral vectors further. This enhanced understanding may allow for the targeted modulation of the immune response to improve vector efficacy and allow for possible redosing. Immune system modulation may involve antibody neutralization, as described in a review of recent research by Herzog and Biswas (2020) [110]. A specific strategy utilizes immunoglobulin-degrading enzymes from Streptococcus that can be administered prior to AAV treatment. The enzymes cut immunoglobulins at a specific site to make them unable to neutralize the vector. This strategy would prevent the development of an immune response, allowing for improved transduction and treatment efficacy [111].

Using viral vectors mandates the co-administration of steroids to prevent transaminitis, a broad immune modification [112]. Although initial study protocols suggested treatment for 30 days followed by a 30-day taper, most patients required steroids longer due to persistent transaminitis. Chand et al. summarized the initial studies with Onasemnogene abeparvovec (Zolgensma) for SMA and noted an average steroid usage of 83 days, ranging from 33 to 229 days [113]. In general, limited use of steroids is safe in infants and children. Steroids are frequently given to neonates with bronchopulmonary dysplasia, infants with infantile spasms, or children with nephrotic syndrome. Common short-term side effects include changes in appetite, mild immunosuppression, and gastrointestinal discomfort. Infants may exhibit changes in hunger or sleep patterns when started on steroids and often have a disrupted vaccination schedule. Stopping steroids after gene transfer becomes more difficult the longer the patient is on the steroids; careful tapering is required to avoid an adrenal crisis. Although common steroids, like prednisone and prednisolone, are relatively affordable compared to gene therapy, the potential side effects from longer-term steroid use could increase the overall cost burden, particularly if hospitalization is required.

## 5. Cost of Gene Therapy

While gene therapy brings significant benefits to patients, it also comes with incredible costs. Research and development have been estimated to cost between USD 318 million and USD 3 billion per gene therapy development [114]. Gene therapy for SMA consists of a one-time intravenous dose. The disease’s rarity ensures a small number of patients receive the medication. The limited usage of the drug drives up the cost. More importantly, this suggests a needed international effort to identify all patients with these rare diseases to reduce cost per patient. Zolgensma, a gene therapy for SMA, costs USD 2.1 million for a one-time dose. The approved gene therapy for hemophilia B, Hemgenix, costs USD 3.5 million per treatment, making it the most expensive drug worldwide, highlighting the need to identify more patients with disease or drug competition to reduce pricing. The high cost of these treatments can be absorbed into the payer’s system because the number of patients requiring treatment is relatively low. This may not be feasible when gene therapy is available for more diseases and a broader population of patients. A cost analysis of gene therapy versus other maintenance therapies for SMA shows that gene therapy is more cost-effective than lifelong intermittent doses of maintenance therapy [115]. This cost-effectiveness is not maintained when SMA patients suffer a relapse [116]. It is also likely to be less cost-effective in more mild diseases. As more data are obtained, cost-effectiveness may not be maintained.

With effective treatments that are more cost-effective for rare diseases, it will be imperative for payment systems to adapt and accommodate the high cost of the medications. It has been suggested that paying smaller amounts over time instead of one large payment before the administration could be an effective mechanism to share the cost between the payers and pharmaceutical companies [112]. It would also ensure the payment could be stopped if the therapy ceases to be effective, similar to stopping the medication if it is no longer effective. This model has already been used in national health plans [117]. Spain and France, for example, will only continue payments for hepatic C treatment if the patient is cured [114]. Novartis also utilizes this approach with Kymriah, a gene therapy for B-cell acute lymphoblastic leukemia. Novartis has an agreement with hospitals that they do not invoice for Kymriah until a 30-day outcome test is completed. No payment is required if the patient does not respond successfully to the treatment in this period [118]. This approach limits the financial burden on patients and hospital systems and increases the financial accessibility of these potentially curative treatments.

In the United States, the Orphan Drug Act (ODA) (1983) was developed to provide financial benefits to pharmaceutical companies for the development of drugs for rare diseases affecting fewer than 200,000 people in the U.S. Some of these benefits include market exclusivity, federal grants, and waivers of marketing application user fees [119]. However, there is a need to incentivize gene therapy development further and reduce the cost of this therapy. These reforms may include implementing a stratified benefit system in which incentives depend on the disease population size and decreasing exclusivity periods to ensure benefits are only utilized for drugs with small patient populations and limited economic potential [119].

## 6. Need for Increased Transparency

Gene therapy has a history of false hope and exaggerated hype. In the early 1990s, completing the first gene therapy clinical trial led to a wave of excitement perpetuated by the media. This enthusiasm spread to researchers and the public alike, leading to the initiation of numerous research projects and a push to advance gene therapy clinical trials. However, this excitement and rapid advancement proved to be self-destructive. In 1995, the NIH released a statement criticizing the field of gene therapy for rushed clinical trials, poor experimental design, and lack of rational scientific logic [120].

This pattern was seen again in 2008, with two reports in the *New England Journal of Medicine* describing a gene therapy to correct a form of congenital blindness. The media extrapolated the results of these reports to suggest the potential for curing eye conditions of all kinds. These statements were met with backlash from the scientific community, specifically about the pressure put on them to accelerate gene therapies [121].

These incidents illustrate how the revolutionary potential of gene therapy needs to be paired with humility. Gene therapy has the potential to do a lot of good, but there are risks and uncertainties. Improved multiway communication between all stakeholders—physicians, researchers, policymakers, companies, patients, and the public—about gene therapy’s risks and benefits is necessary. The information conveyed to patients and the general public should be clear, relatable, concise, and reliable. This information may be paired with increased genetic education through genetic counseling for patients and their families, as knowledge of genetics is crucial to understanding gene therapy’s risks and benefits [122].

The potential for side effects, the possibility that effectiveness may wane, and the plethora of new gene therapy drugs in the pipeline necessitate discussion between researchers, clinicians, and patients. This ongoing discussion will be essential to ensure side effects are noted swiftly, and changes to clinical practice can be made. Currently, rare disease advocacy groups have well-established registries collecting patient data across institutions, including groups serving multiple diagnoses like the Muscular Dystrophy Association and groups specific to one disease process like CureSMA, CureDuchenne, and Parent Project Muscular Dystrophy. These databases have years of patient information and already have the infrastructure to collect information on safety and patient outcomes as gene therapy is used and implemented in the future. These groups serve as a valuable resource for communication between patients, clinicians, and researchers.

Physicians from every specialty should know about the field to effectively communicate relevant information to their patients. Physicians and researchers should work together to ensure access to relevant information about current gene therapy developments to keep patients well informed about the current state of research. However, not all education is top-down. Researchers also need to hear from patients about their concerns and experiences to ensure research efforts align with the needs of the patient population for which they are developing treatments [120].

The high cost of gene therapy leads to a complicated pay structure. This requires clinicians, payers, and hospital systems to communicate to ensure timely patient drug delivery. Lastly, communication between policymakers, clinicians, patients, payers, and hospital systems must be prioritized to ensure safety and equitable distribution are established.

An increase in information sharing between companies and researchers, specifically about failed clinical trials, is also imperative to the informational accessibility of gene therapy. After a failed phase III clinical trial for gene therapy for epidermolysis bullosa, the company leading the study contacted other companies working on the disease and unpacked their data, presenting what they had learned from the failed study. As a result, one of the companies changed its inclusion endpoints [123]. This model of accountability and transparency is necessary for the future progression of gene therapy. The success of a gene therapy clinical trial hinges on multiple components, such as the vector selection, the gene delivered, and the promoter utilized. The accessibility of this information is essential to the analysis of both prior and present clinical trials to analyze current trends in trial design and common denominators for observed outcomes.

## 7. Ethical Considerations for Gene Therapy—Conclusions

The ethics of gene therapy are as multi-faceted as the field of medicine itself. We have laid out the biological, clinical, and public/patient-centric ethical considerations of gene therapy within this article (Figure 10). However, the ethical issues surrounding gene therapy are less about gene therapy itself and more about the medical, cultural, social, and political contexts in which it emerged. We cannot boil down these questions and issues to one-time decisions and solutions, which would disregard the relational and longitudinal nature of ethics [124]. Addison and Lassen unravel the concept of the ethics of gene therapy clinical trials as follows: “The ethical complexities of gene therapy are not confined to the consent process or the procedure, nor does the ethics review process resolve them. Rather, the treatment unfurls a multitude of ethical dilemmas, which manifest both in discrete moments of choice and the on-going endeavor of how to live well or care well in the aftermath of the event itself”.

The Hippocratic Oath [125], often referred to as the basis of ethical medical practice, presents the purpose of medicine as “to do away with suffering of the sick, to lessen the violence of their diseases.” The purpose of medicine and the principle of ethical practice hinge on relieving patient suffering, which, at its core, is patient-centered [126]. Therefore, the ethical advancement of gene therapy hinges on developing patient-centered solutions to the present and emerging ethical dilemmas and issues faced within this field. With every decision and every advancement, we must remember the patient.

This patient-centered lens can serve as the basis for thinking about the ethics of many gene therapy topics we have discussed. As evident in our analysis of gene therapy clinical trials, gene therapy is still in its early stages of development, with most clinical trials falling into the early phase categories (phases I, I/II, and II). The lack of international partnerships has prevented the scale of gene therapy from matching the rarity of diseases it is being developed to treat, representing a significant ethical consideration for cross-border study designs [19].

Severe adverse events, even patient deaths, although they are to be actively avoided through proper monitoring and reporting, are not uncommon within early phase trials, especially phase I trials [127]. In 1999, 153,964 severe adverse events (17,399 of them patient deaths) were reported to the Center for Drug Evaluation and Research of the United States FDA [128]. That same year, the phase I gene therapy clinical trial for OTC deficiency resulting in death was highly publicized, with 22 *New York Times* articles [129]. The media focusses on gene therapy more than other disciplines, leading to an amplified perception of risk. We must be clear about who these risks fall upon. Ultimately, they fall upon the patients—those actively involved in trials, those who will receive these treatments in the future, and those directly and indirectly affected by the outcomes of these discussions and decisions. Therefore, we must actively involve patient populations in the discussions and decision-making processes about the acceptable level of risk.

One option discussed by Pattee in their commentary titled “Protections for Participants in Gene Therapy Trials: A Patient’s Perspective” [130] is to consult patients who have participated in trials on trial design, development, and direction, such as ensuring the adequacy of informed consent materials and trial logistics. Doing so would increase trial transparency and public trust in gene therapy, even amid complex uncertainties within the field. Pattee also suggests further protecting patients participating in clinical trials through improved public education about clinical trials to clarify information and concerns presented in the media and including disease-specific experts within centralized IRBs to incorporate additional perspectives specific to the patient population during trial design and monitoring [130].

Accessibility is a crucial factor to be considered in the ethical advancement of gene therapy. Rare diseases affect a small number of individuals in distinct ways. No two patients are identical. Gene therapy reflects the patient population in this way—it is designed to be specialized. The needs are not equal; therefore, treatments cannot be equal. However, treatment equity is still needed, from costs to the type of rare disease to trial access that disproportionately benefits a few [118,131]. To think about equity and accessibility is to consider already present disparities in healthcare systems, patterns we see emerging from early research and clinical trials, and other potential barriers that could threaten the ethical advancement of gene therapy, the safety of patient populations, and the ability of patients to access these potentially curative treatments.

Over 50% of individuals with rare diseases report using their savings to cover medical costs, with one in ten filing for bankruptcy [123]. The high cost of gene therapies is thus likely to continue overwhelming patients with rare diseases and the funding agencies for medical care, thus limiting personal access. As shown in Table 1, Table 2 and Table 3, only a few rare diseases have authorized gene therapies, where the >5000 unique rare diseases represent a significant opportunity to reduce production costs through transparent design that enables the subsequent therapy to be developed at a lower cost. Further expansion of international collaborations will unite rare disease patients to present a more extensive base of therapies. No matter how effective or miraculous, a treatment inaccessible to patients has no real value. Thus, a balance of patient risk, education, and accessibility remains the ethical priority for gene therapy of rare diseases.

## Figures and Tables

**Figure 1 biotech-13-00001-f001:**
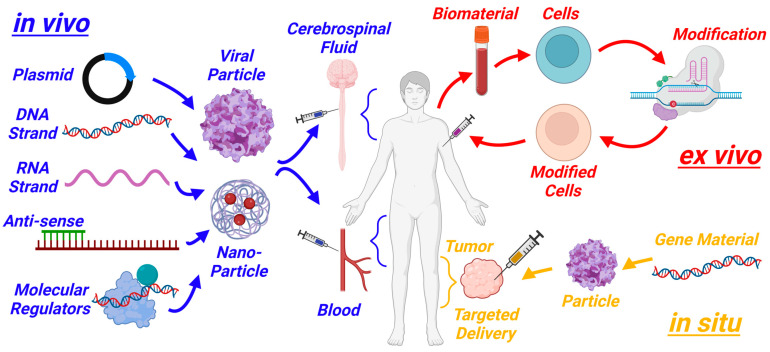
Schematic of three gene therapy approaches: in vivo, ex vivo, and in situ. Generated with BioRender.

**Figure 2 biotech-13-00001-f002:**
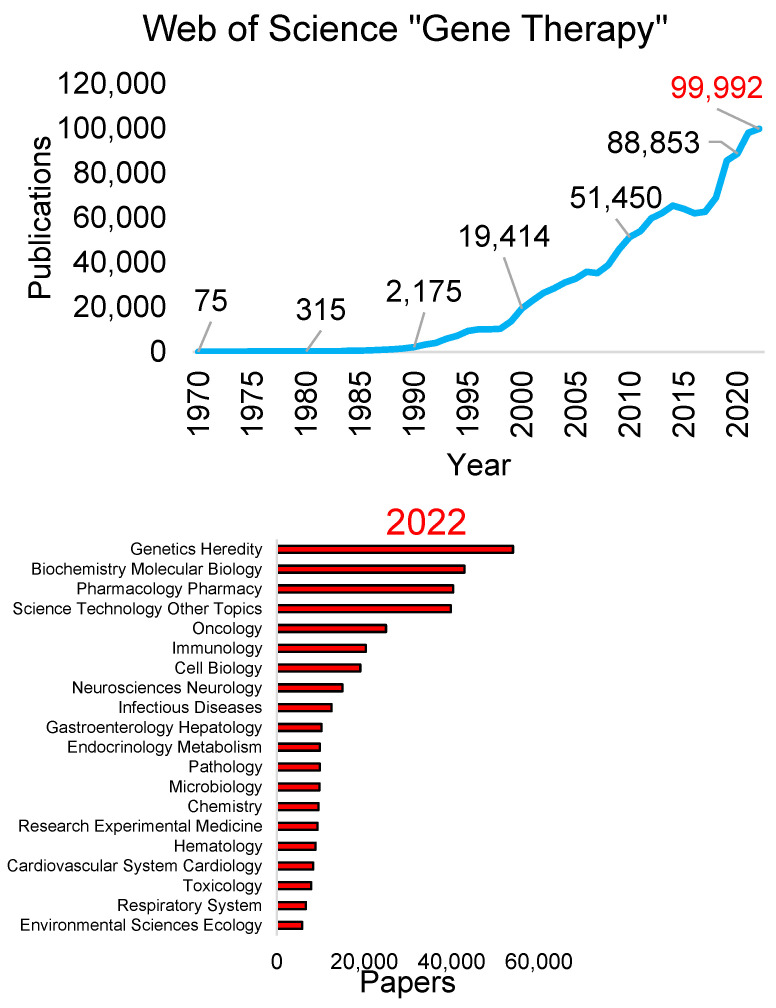
Publications on “gene therapy.” The first panel shows the number of publications found on Web of Science per year for the search “gene therapy,” with every five years labeled in black. The number of publications in 2022 is in red. The second panel shows the breakdown of the top 20 research areas of the 2022 papers. The analysis was performed on 18 April 2023.

**Figure 3 biotech-13-00001-f003:**
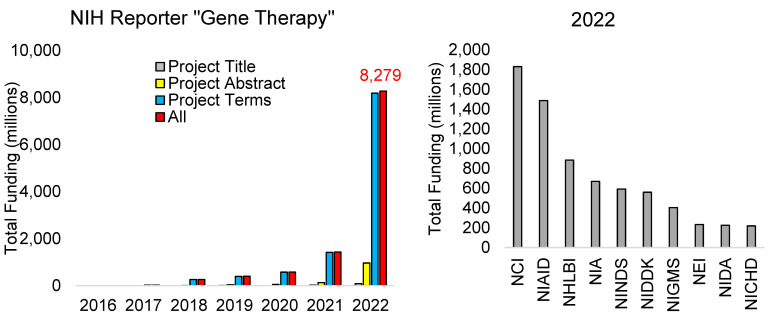
NIH funding mentioning “Gene Therapy.” The first panel shows the funding (in millions of USD) per year by the National Institutes of Health (NIH) mentioning the term “gene therapy” in various annotation bins (mentioned in project: gray—title, yellow—abstract, cyan—terms, red—any of the three). The total annotated funding in 2022 is in red text. The second panel shows the breakdown of the top NIH institutes of the 2022 NIH funding for “Gene Therapy.” Abbreviations: NCI—National Cancer Institute, NIAID—National Institute of Allergy and Infectious Diseases, NHLBI—National Heart, Lung, and Blood Institute, NIA—National Institute on Aging, NINDS—National Institute of Neurological Disorders and Stroke, NIDDK—National Institute of Diabetes and Digestive and Kidney Diseases, NIGMS—National Institute of General Medical Sciences, NEI—National Eye Institute, NIDA—National Institute on Drug Abuse, NICHD—Eunice Kennedy Shriver National Institute of Child Health and Human Development. The analysis was performed on 1 May 2023 using NIH reporter.

**Figure 4 biotech-13-00001-f004:**
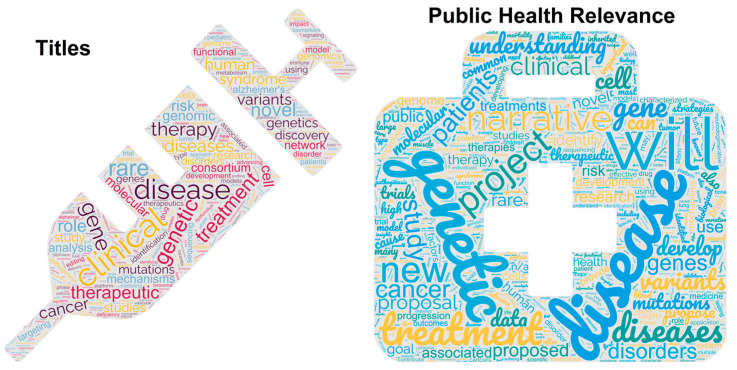
Word usage for “gene therapy” and “rare disease” within all NIH-funded projects. The analysis was performed using WordClouds.com. The first panel shows words enriched within the 787 funded project titles. The second panel shows words enriched from their public health relevance statements. The analysis was performed on 1 May 2023 using NIH reporter.

**Figure 5 biotech-13-00001-f005:**
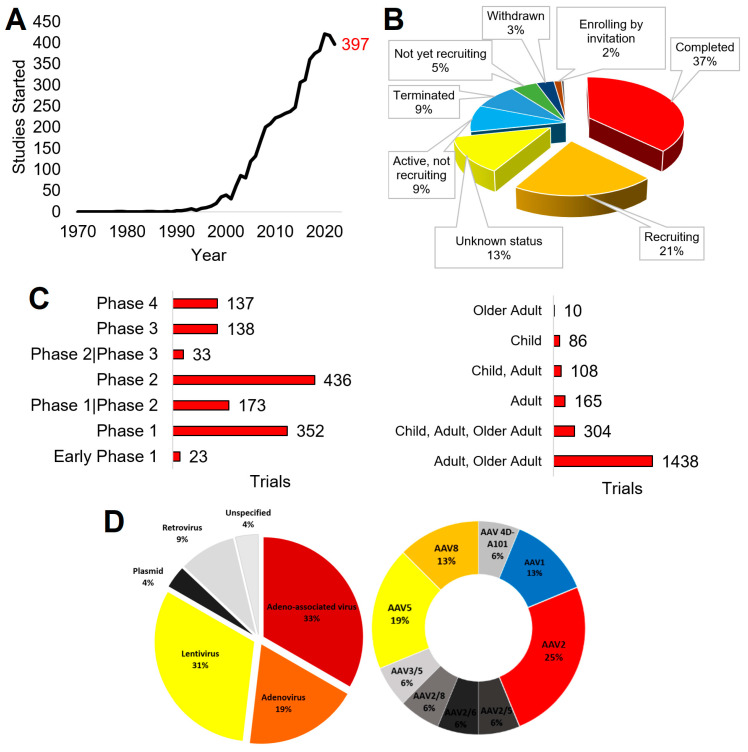
Analysis of ClinicalTrials.gov for “gene therapy.” All analyses were performed on 18 April 2023 using the ClinicalTrial.gov site. (**A**) Number of trials started each year, with the 2022 number in red. (**B**) Breakdown of trial status. Groups below 2% are not shown. (**C**) Breakdown of completed trials for FDA phase and age group inclusion. (**D**) Breakdown of the delivery system used, with a call out of adeno-associated virus subtypes shown to the right.

**Figure 6 biotech-13-00001-f006:**
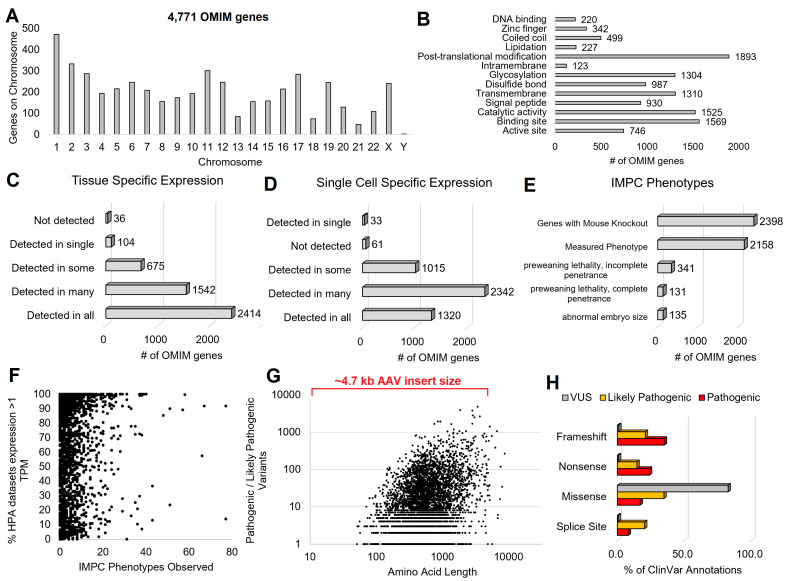
OMIM genes connecting human genotypes to phenotypes. (**A**) Number of OMIM genes per chromosome. (**B**) The number of OMIM genes with various human UniProt annotations. (**C**) Tissue- or (**D**) single-cell-specific expression annotation from the Human Protein Atlas for each of the OMIM genes. (**E**) The number of OMIM genes with various International Mouse Phenotyping Consortium (IMPC) annotations following knockout and phenotyping. (**F**) Each OMIM gene number of IMPC phenotypes altered in knockout (x-axis) relative to the % of datasets from the Human Protein Atlas where the gene is expressed >1 transcript per million (TPM). (**G**) The amino acid length of each OMIM gene (x-axis) relative to the number of ClinVar annotated pathogenic or likely pathogenic variants. (**H**) The percent of each variant class relative to variant alterations for the ClinVar database.

**Figure 7 biotech-13-00001-f007:**
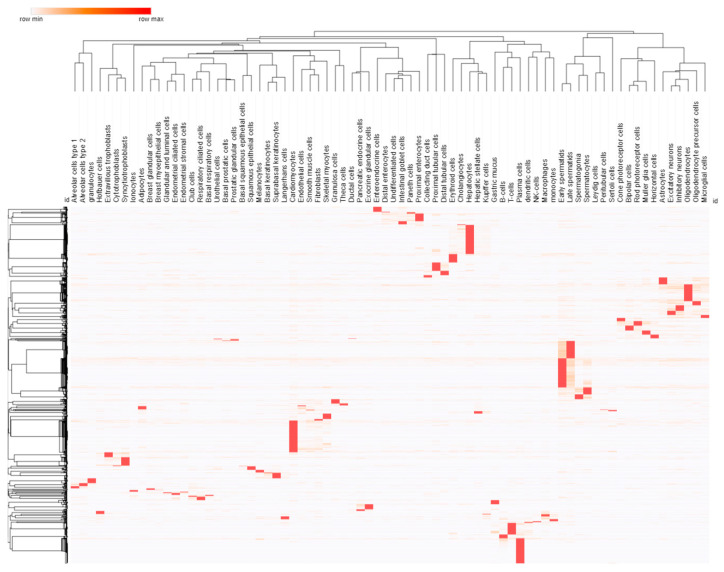
Heatmap of expression for cell-type-specific genes from the Human Protein Atlas. Red indicates the highest expression in the row. Dendrograms show one minus Spearman rank correlation with cell type on top and genes shown to the left.

**Figure 8 biotech-13-00001-f008:**
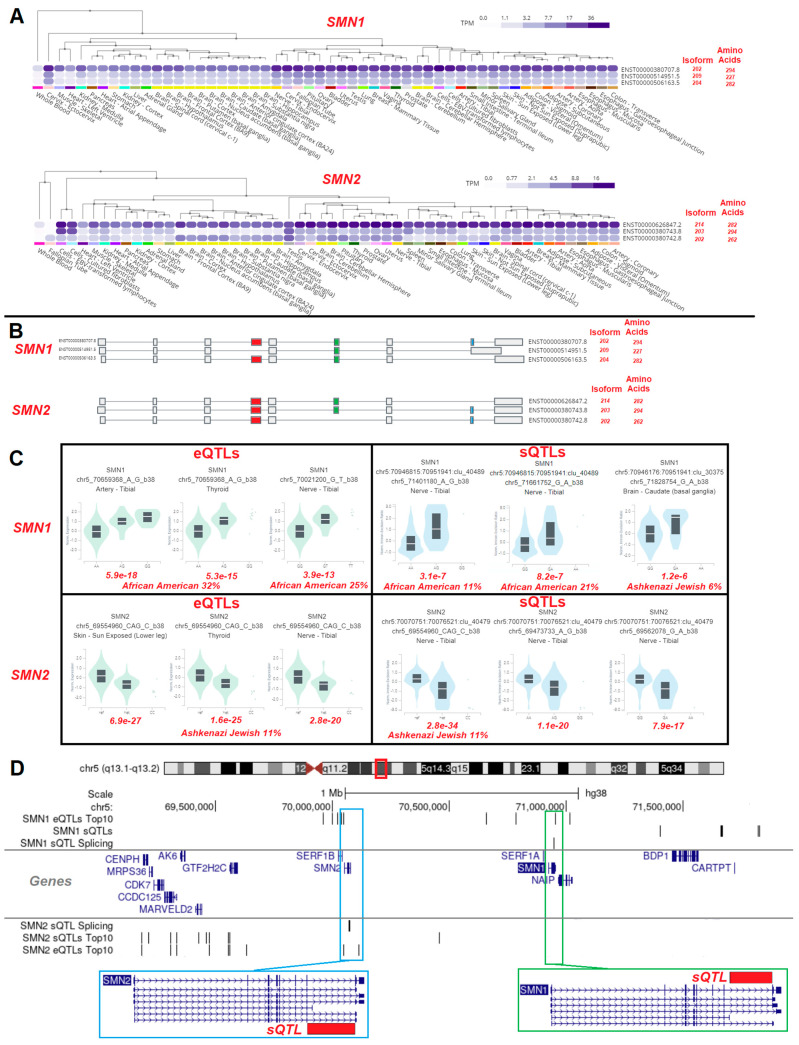
Isoforms and genetics of SMN1 and SMN2. (**A**) Top three protein-coding isoforms for SMN1 and SMN2 genes. (**B**) Exon map of isoforms within panel (**A**). (**C**) GTEx-measured eQTLs and sQTLs for the SMN1 and SMN2 genes. The significance and the population with the highest frequency of the variants are labeled in red below the violin plots. (**D**) Chromosome 5 map of the top eQTL and sQTL signals for SMN1 and SMN2.

**Figure 9 biotech-13-00001-f009:**
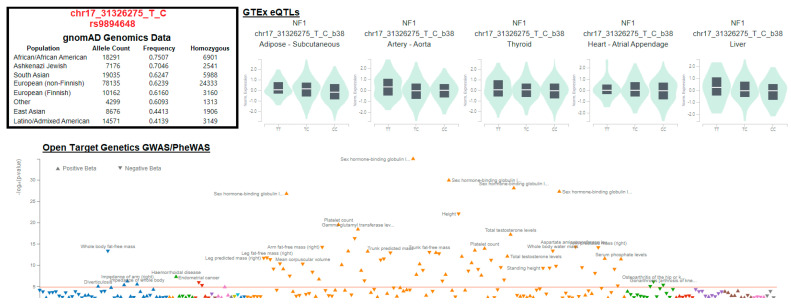
Representative analysis of a variant colocalized for expression and phenotypes. The first panel shows variant allele frequency data from gnomAD population genomics sequencing. The GTEx eQTL plots show five different tissues with significant eQTLs for the variant within the NF1 gene. The bottom plot shows the Open Target Genetics [76] data curation for significant traits associated with this variant.

**Figure 10 biotech-13-00001-f010:**
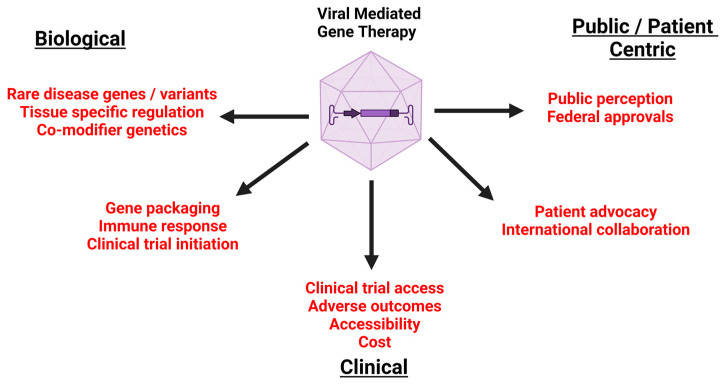
Summary of the ethical considerations of gene therapy. This figure was generated with BioRender.

**Table 1 biotech-13-00001-t001:** Top ten highest NIH-funded projects mentioning “gene therapy”. The analysis was performed on 1 May 2023 using NIH reporter.

Application ID	Project Number	Total Cost I.C.	Administering I.C.	Organization Name	Project Title
10695742	1ZIATR000437-01	USD 77,500,000	NCATS	National Center for Advancing Translational Sciences	Antiviral Program for Pandemics (App) and Ncats: Accelerating Antiviral Development
10514264	1U19AI171421-01	USD 69,058,677	NIAID	Stanford University	Development of Outpatient Antiviral Cocktails Against SARS-CoV-2 and Other Potential Pandemic Rna Viruses.
10514317	1U19AI171443-01	USD 67,624,156	NIAID	Scripps Research Institute, The	Center For Antiviral Medicines and Pandemic Preparedness (Campp)
10512617	1U19AI171110-01	USD 67,452,049	NIAID	University of California, San Francisco	Qcrg Pandemic Response Program
10522804	1U19AI171954-01	USD 66,431,207	NIAID	University of Minnesota	Midwest Avidd Center
10513679	1U19AI171292-01	USD 65,483,194	NIAID	Univ of North Carolina Chapel Hill	Rapidly Emerging Antiviral Drug Development Initiative—Avidd Center (Readdi-Ac)
10513935	1U19AI171403-01	USD 51,914,880	NIAID	Emory University	Antiviral Countermeasures Development Center (Ac/Dc)
10716676	75N91019D00024-0-759102200019-1	USD 22,364,766	NCI	Leidos Biomedical Research, Inc.	Discovery and Development of Cancer Therapeutics for Next Program
10446989	5U01AG059798-03	USD 20,263,304	NIA	Washington University	Dian-Tu Primary Prevention Trial
10649756	1UF1NS131791-01	USD 18,136,504	NINDS	Massachusetts General Hospital	An Expanded Access Protocol of Intravenous Trehalose Injection 90 mg/mL Treatment of Patients with Amyotrophic Lateral Sclerosis
10693707	1ZIAHD002400-31	USD 17,942,380	NICHD	Eunice Kennedy Shriver National Institute of Child Health and Human Development	The Role of Subclinical Infection and Cytokines in Preterm Parturition
10452692	5R01AG068319-03	USD 16,720,909	NIA	Washington University	Dian-Tu: Tau Next Generation Prevention Trial
9457012	1U24CA224319-01	USD 13,559,983	NCI	Icahn School of Medicine at Mount Sinai	High-Dimensional Immune Monitoring of Nci-Supported Immunotherapy Trials
10266149	5U19NS120384-02	USD 13,212,214	NINDS	University Of California at Davis	The Clinical Significance of Incidental White Matter Lesions on Mri Amongst a Diverse Population with Cognitive Complaints (Indeed)

**Table 2 biotech-13-00001-t002:** Top genetic diseases with interventional “gene therapy” clinical trials. A “-“ is used in the FDA-authorized treatment column when no treatments are authorized. The “*” indicates drugs that are not gene therapy.

Disorder	Trials	Total Enrollment	Trials with under 18	Phase I	Phase I |Phase II	Phase II	Phase II |Phase III	Phase III	FDA-Authorized Treatment
Cystic Fibrosis	43	4080	27	7	3	15	2	10	Elexacaftor–Tezacaftor–Ivacaftor *
Hemophilia B	26	666	2	5	11	4	0	3	Hemgenix
Retinal Dystrophy	2	35	2	0	0	0	0	2	Luxturna
Cerebral Adrenoleukodystrophy	2	67	2	0	0	0	1	1	Skysona
Spinal Muscular Atrophy	14	713	13	2	1	0	0	8	Zolgensma
β-Thalassemia	25	604	17	2	7	3	1	2	Zynteglo
Muscular Dystrophy	42	1837	37	9	12	12	0	7	Elevidys
Hemophilia A	22	678	1	5	7	2	0	5	Roctavian
Epidermolysis Bullosa	18	228	15	0	12	2	0	2	Vyjuvek
Fabry Disease	12	377	5	1	7	1	0	1	-
Sickle Cell Anemia	5	245	2	0	0	2	0	1	-
Mucopolysaccharidosis	17	186	17	1	13	0	2	0	-
Gaucher Disease	13	366	8	0	6	0	2	5	-
Retinitis Pigmentosa	27	1713	13	1	14	3	3	3	-
Leber Congenital Amaurosis	11	178	11	2	6	0	2	1	-
Amyotrophic Lateral Sclerosis	5	308	0	0	1	2	0	1	-
Severe Combined Immunodeficiency	19	181	19	3	11	1	1	0	-
Fanconi Anemia	14	82	14	6	4	3	0	0	-
Alzheimer’s Disease	4	43	0	3	1	0	0	0	-

**Table 3 biotech-13-00001-t003:** Curated phase III intervention studies for genetic syndromes.

Trial	Status	Phases	Start Date	Completion Date	Age	Enrollment #	Conditions	Interventions
NCT02292537	Completed	Phase III	24 November 2014	20 February 2017	2–12 years	126	Spinal Muscular Atrophy	Nusinersen (Spinraza)
NCT03306277	Completed	Phase III	24 October 2017	12 November 2019	up to 180 Days	22	Spinal Muscular Atrophy	Biological: Onasemnogene Abeparvovec
NCT03461289	Completed	Phase III	16 August 2018	11 September 2020	up to 6 Months	33	Spinal Muscular Atrophy	Biological: Onasemnogene Abeparvovec
NCT03496012	Completed	Phase III	11 December 2017	1 December 2020	18 Years and older	170	Choroideremia	Genetic: BIIB111
NCT01896102	Completed	Phase III Phase III	21 August 2013	26 March 2021	up to 17 Years	32	Cerebral Adrenoleukodystrophy (CALD)	Genetic: Lenti-D Drug Product
NCT03505099	Completed	Phase III	2 April 2018	15 June 2021	up to 42 Days	30	Spinal Muscular Atrophy	Biological: Onasemnogene Abeparvovec
NCT03837184	Completed	Phase III	31 May 2019	29 June 2021	0 Days to 6 Months	2	Spinal Muscular Atrophy	Biological: Onasemnogene Abeparvovec
NCT02906202	Completed	Phase III	1 July 2016	31 March 2022	0 Years to 50 Years	23	β-Thalassemia	Genetic: LentiGlobin BB305
NCT03406104	Completed	Phase III	9 January 2018	4 July 2022	15 Years and older	61	Leber Hereditary Optic Neuropathy	Genetic: GS010
NCT03207009	Completed	Phase III	8 June 2017	15 November 2022	0 Years to 50 Years	18	β-Thalassemia	Genetic: LentiGlobin BB305
NCT00999609	Active, not recruiting	Phase III	1 October 12	-	3 Years and older	31	Inherited Retinal Dystrophy	Biological: AAV2-hRPE65v2
NCT03370913	Active, not recruiting	Phase III	19 December 2017	-	18 Years and older	134	Hemophilia A	Biological: valoctocogene roxaparvovec
NCT03293524	Active, not recruiting	Phase III	12 March 2018	-	15 Years and older	90	Leber Hereditary Optic Neuropathy	Genetic: GS010
NCT03392974	Active, not recruiting	Phase III	14 March 2018	-	18 Years and older	1	Hemophilia A	Biological: Valoctocogene Roxaparvovec
NCT03569891	Active, not recruiting	Phase III	27 June 2018	-	18 Years and older	67	Hemophilia B	Genetic: AAV5-hFIXco-Padua (Hemgenix)
NCT03837483	Active, not recruiting	Phase III	21 January 2019	-	up to 65 Years	10	Wiskott–Aldrich Syndrome	Genetic: OTL-103
NCT03852498	Active, not recruiting	Phase III	24 January 2019	-	up to 17 Years	35	Cerebral Adrenoleukodystrophy (CALD)	Genetic: Lenti-D
NCT04042025	Active, not recruiting	Phase III	10 February 2020	-	Child, Adult, Older Adult	85	Spinal Muscular Atrophy	Biological: Onasemnogene Abeparvovec
NCT04323098	Active, not recruiting	Phase III	10 November 2020	-	18 Years and older	22	Hemophilia A	Biological: valoctocogene roxaparvovec
NCT04516369	Active, not recruiting	Phase III	24 November 2020	-	4 Years and older	4	Retinal Dystrophy	Genetic: voretigene neparvovec (LUXTURNA)
NCT04851873	Active, not recruiting	Phase III	8 September 2021	-	up to 17 Years	24	Spinal Muscular Atrophy	Genetic: OAV101
NCT05096221	Active, not recruiting	Phase III	27 October 2021	-	4 Years to 7 Years	126	Duchenne Muscular Dystrophy	Genetic: SRP-9001
NCT05139316	Active, not recruiting	Phase III	8 November 2021	-	8 Years and older	50	Glycogen Storage Disease Type IA	Genetic: DTX401
NCT03566043	Recruiting	Phase II| Phase III	27 September 2018	-	4 Months to 5 Years	48	Mucopolysaccharidosis Type II (MPS II)	Genetic: RGX-121
NCT03861273	Recruiting	Phase III	29 July 2019	-	18 Years to 65 Years	55	Hemophilia B	Biological: fidanacogene elaparvovec
NCT04293185	Recruiting	Phase III	14 February 2020	-	2 Years to 50 Years	35	Sickle Cell Disease	Genetic: bb1111
NCT04370054	Recruiting	Phase III	18 August 2020	-	18 Years to 64 Years	63	Hemophilia A	Biological: PF-07055480
NCT04281485	Recruiting	Phase III	5 November 2020	-	4 Years to 7 Years	99	Duchenne Muscular Dystrophy	Genetic: PF-06939926
NCT04704921	Recruiting	Phase II| Phase III	29 December 2020	-	50 Years to 89 Years	300	Age-related Macular Degeneration	Genetic: RGX-314
NCT04671433	Recruiting	Phase III	16 March 2021	-	3 Years and older	96	X-Linked Retinitis Pigmentosa	Genetic: AAV5-RPGR
NCT04794101	Recruiting	Phase III	16 March 2021	-	3 Years and older	96	X-Linked Retinitis Pigmentosa	Genetic: AAV5-RPGR
NCT05407636	Recruiting	Phase III	28 December 2021	-	50 Years to 89 Years	465	Age-related Macular Degeneration	Genetic: RGX-314
NCT05089656	Recruiting	Phase III	12 January 2022	-	2 Years to 17 Years	125	Spinal Muscular Atrophy	Genetic: OAV101
NCT04283227	Recruiting	Phase III	17 January 2022	-	Child, Adult, Older Adult	6	Lysosomal Storage Diseases	Genetic: OTL-200
NCT05345171	Recruiting	Phase III	18 October 2022	-	12 Years and older	50	OTC Deficiency	Genetic: DTX301
NCT05335876	Recruiting	Phase III	19 December 2022	-	Child, Adult, Older Adult	260	Spinal Muscular Atrophy	Biological: onasemnogene abeparvovec
NCT05386680	Recruiting	Phase III	12 January 2023	-	2 Years to 12 Years	28	Spinal Muscular Atrophy	Genetic: OAV101
NCT05689164	Not yet recruiting	Phase III	14 April 2023	-	0 Years and older	250	Duchenne Muscular Dystrophy	Biological: fordadistrogene movaparvovec
NCT05815004	Not yet recruiting	Phase II| Phase III	1 October 2023	-	2 Years to 25 Years	40	Gaucher Disease, Type 3	Drug: Gene therapy
NCT00073463	Terminated	Phase II| Phase III	1 June 2003	-	12 Years and older	100	Cystic Fibrosis	Genetic: tgAAVCF

## Data Availability

Not applicable.

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
