# Peer review of "Gene Therapy for Genetic Syndromes: Understanding the Current State to Guide Future Care"

_biotech, 2024, doi:10.3390/biotech13010001_

Round 1
Reviewer 1 Report
Comments and Suggestions for Authors
Authors proposed a paper entitled “Gene Therapy for Genetic Syndromes: Understanding the Current State to Guide Future Care” for the publication in BioTech”.
The paper has a good scientific soundness, but some parts need to be clarified or improved.
Authors are invited to add an Abbreviation list, according to the guidelines of this journal.
This paper also contains a quite complete state of the art study; however, it should be improved with the addition of references as support.
Here are some issues and opinions line by line:
Line 68. Information about rare diseases is given only for US country. Is there the possibility to add information also about other countries or continents, in order to make comparison and give a global estimation?
Line 76. “CRISPR” please define it.
Lines 80-82. These two lines are where the authors express directly what their goals are in this work. I suggest expanding this in order to clarify their intention with this review paper.
Line 121. “so we are not blinded” I would choose more impersonal expressions.
Line 158. I would say “among these awards…” instead of “of”.
Table 1. please compact this table according to the guidelines of this journal.
Line 236. Same issue of Line 158.
Table 2. please compact the first column about disorders and clarify if the empty cells of the last column are not authorized from FDA.
Table 3 needs to be compacted and synthetized.
Line 296. Bold font is not needed in these paragraphs.
Line 422. “identify when gene therapy will be safe and effective” can you add reference here to confirm your affirmation?
Line 455. “Of the OMIM genes” same issue of Line 158.
Line 458. “that human variants can result in altered splicing.” Please add a reference here.
Line 558. “gene therapy would only require one dose.” Please clarify this affirmation adding explanation.
Line 559. “long-term data on the efficacy of these treatments” can you confirm this with literature references?
Line 573. “Jesse Gelsinger” did you get the permission to publish the name of this patient?
Line 635. “Research and development” is this cost defined on annual basis? Or are you intending one-time dose cost, as indicated in line 639 ?
Line 640. “making it the most expensive drug worldwide” are there any strong competitors than can guarantee same positive effects and lower prices worldwide?
Even if the last paragraph contains conclusive remarks, authors should add a “conclusions” paragraph in this paper.
Comments on the Quality of English LanguageA quite good use of English. However I suggest performing a light revision of the manuscript.
Author Response
The paper has a good scientific soundness, but some parts need to be clarified or improved.
Response: We thank reviewer 1 for the very thoughtful revision requests. We have gone through line by line of the comments with responses.
Authors are invited to add an Abbreviation list, according to the guidelines of this journal.
Response: We have added this at the end of the paper.
This paper also contains a quite complete state of the art study; however, it should be improved with the addition of references as support.
Response: We have added additional references throughout the article to further support statements.
Line 68. Information about rare diseases is given only for US country. Is there the possibility to add information also about other countries or continents, in order to make comparison and give a global estimation?
Response: We have added half a paragraph and citations on the international rare disease numbers, the importance for international collaboration, and ongoing initiatives to translate data across borders.
Line 76. “CRISPR” please define it.
Response: Added at first use and in the abbreviation list placed at the end of the article.
Lines 80-82. These two lines are where the authors express directly what their goals are in this work. I suggest expanding this in order to clarify their intention with this review paper.
Response: We have added three sentences that address the sections of the article.
Line 121. “so we are not blinded” I would choose more impersonal expressions.
Response: Changed to “These findings highlight the persistent need for refined knowledge of how foreign nucleotides can impact cellular processes to better predict unexpected, off target outcomes.”
Line 158. I would say “among these awards…” instead of “of”.
Response: Changed as suggested.
Table 1. please compact this table according to the guidelines of this journal.
Response: We have updated the table into the format of the word template.
Line 236. Same issue of Line 158.
Response: Changed as suggested.
Table 2. please compact the first column about disorders and clarify if the empty cells of the last column are not authorized from FDA.
Response: We added a “-“ for any without authorization. We updated the table format according to the word template. We also added three new authorized treatments that were approved.
Table 3 needs to be compacted and synthetized.
Response: The table is based on unique trial IDs and thus we are unsure how to compact further without loosing resolution of unique trials. We welcome suggestions for further refinement.
Line 296. Bold font is not needed in these paragraphs.
Response: There was no bold font used, just all capital letters of the drugs and links to the FDA pages, which have an appearance of being bold.
Line 422. “identify when gene therapy will be safe and effective” can you add reference here to confirm your affirmation?
Response: We added two reference for this statement.
Line 455. “Of the OMIM genes” same issue of Line 158.
Response: Corrected.
Line 458. “that human variants can result in altered splicing.” Please add a reference here.
Response: We added a citation to GTEx.
Line 558. “gene therapy would only require one dose.” Please clarify this affirmation adding explanation.
Response: Changed to “Currently, gene therapy is designed to deliver the desired effect in one dose.”
Line 559. “long-term data on the efficacy of these treatments” can you confirm this with literature references?
Response: Changed to “However, there is a lack of long-term data on the efficacy of these treatments as the FDA approvals have only been in the past few years” and added a reference.
Line 573. “Jesse Gelsinger” did you get the permission to publish the name of this patient?
Response: While this is common knowledge of the name and the case, we reframed the reference to remove the name.
Line 635. “Research and development” is this cost defined on annual basis? Or are you intending one-time dose cost, as indicated in line 639 ?
Response: Changed to “Research and development have been estimated to cost between $318 million to $3 bil-lion per gene therapy development.”
Line 640. “making it the most expensive drug worldwide” are there any strong competitors than can guarantee same positive effects and lower prices worldwide?
Response: This is a great point. We added, “highlighting the need for identifying more patients with disease or drug competition to reduce pricing.”
Even if the last paragraph contains conclusive remarks, authors should add a “conclusions” paragraph in this paper.
Response: We have added a section 7 “Ethical Considerations for Gene Therapy - Conclusion” based on the editorial feedback on ethics. This new section aims to sum the ethical considerations, the focus of the paper call we responded to for this article.
Reviewer 2 Report
Comments and Suggestions for Authors
This is a very good review article. It presents the current state while laying forth tools and resources to guide informed directions to avoid foreseeable issues in gene therapy that could prevent the field from continued success. I recommend publication of this work in BioTech.
This manuscript well summarized important facts especially with good figures. However, the presented figures are rather complicated ones. It may not be good for understanding of non-specialist readers. Therefore, I recommend the authors to add one final figure to summarize general images on the Current State to Guide Future Care.
Author Response
This manuscript well summarized important facts especially with good figures. However, the presented figures are rather complicated ones. It may not be good for understanding of non-specialist readers. Therefore, I recommend the authors to add one final figure to summarize general images on the Current State to Guide Future Care.
Response: We have added a new summary figure for the ethical considerations of gene therapy into the new last section with hopes to summarize the manuscripts take away points. We thank the reviewer for this outstanding idea.
Reviewer 3 Report
Comments and Suggestions for Authors
The manuscript represents a detailed overview of gene therapy from the use, the cost, and part of the rationale. It is well-written, and it is excellent for students and professionals with interest in the issue, but no expertise. I recommend it for publication
Comments on the Quality of English LanguageMinor grammatical mistakes were observed.
Author Response
The manuscript represents a detailed overview of gene therapy from the use, the cost, and part of the rationale. It is well-written, and it is excellent for students and professionals with interest in the issue, but no expertise. I recommend it for publication.
Response: We thank the reviewer for the kind review. The manuscript has been edited with several additions to summarize insights for the general reader and for the minor grammatical mistakes. We hope these revisions increase the utility of the article.
Round 2
Reviewer 1 Report
Comments and Suggestions for Authors
Authors proposed a revised version of their paper, after responded point by point to my issues.
The abbreviation list has been inserted at Line 157 and at Line 826. Why authors choose this way? I would suggest adding the abbreviation list in an unique paragraph at the end of the paper, before references section.
Line 49. “CRISPR” there is not an opening parenthesis, but a symbol “/”. Please change it with the symbol "("
Author’s goals and intentions were expanded as requested.
Table 1. I would try to improve the use of space in these lines. Some words do not even fill a line in a single cell.
Author Response
The abbreviation list has been inserted at Line 157 and at Line 826. Why authors choose this way? I would suggest adding the abbreviation list in an unique paragraph at the end of the paper, before references section.
Response: Abbreviations moved to the end of paper just before references.
Line 49. “CRISPR” there is not an opening parenthesis, but a symbol “/”. Please change it with the symbol "("
Response: corrected
Table 1. I would try to improve the use of space in these lines. Some words do not even fill a line in a single cell.
Response: We have adjusted table spacing